# In Vitro Disease Models of the Endocrine Pancreas

**DOI:** 10.3390/biomedicines9101415

**Published:** 2021-10-08

**Authors:** Marko Milojević, Jan Rožanc, Jernej Vajda, Laura Činč Ćurić, Eva Paradiž, Andraž Stožer, Uroš Maver, Boštjan Vihar

**Affiliations:** 1Faculty of Medicine, Institute of Biomedical Sciences, University of Maribor, Taborska ulica 8, 2000 Maribor, Slovenia; jan.rozanc@um.si (J.R.); jernej.vajda1@um.si (J.V.); laura.cinc@um.si (L.Č.Ć.); uros.maver@um.si (U.M.); 2Department of Pharmacology, Faculty of Medicine, University of Maribor, Taborska ulica 8, 2000 Maribor, Slovenia; 3Faculty of Medicine, Institute of Physiology, University of Maribor, Taborska ulica 8, 2000 Maribor, Slovenia; eva.paradiz1@um.si (E.P.); andraz.stozer@um.si (A.S.); 4Institute for Development of Advanced Applied Systems—IRNAS Ltd., Limbuška cesta 76b, 2000 Maribor, Slovenia

**Keywords:** in vitro disease models, pancreas, islet of Langerhans, 3D cell culture, scaffolds, acute tissue slices

## Abstract

The ethical constraints and shortcomings of animal models, combined with the demand to study disease pathogenesis under controlled conditions, are giving rise to a new field at the interface of tissue engineering and pathophysiology, which focuses on the development of in vitro models of disease. In vitro models are defined as synthetic experimental systems that contain living human cells and mimic tissue- and organ-level physiology in vitro by taking advantage of recent advances in tissue engineering and microfabrication. This review provides an overview of in vitro models and focuses specifically on in vitro disease models of the endocrine pancreas and diabetes. First, we briefly review the anatomy, physiology, and pathophysiology of the human pancreas, with an emphasis on islets of Langerhans and beta cell dysfunction. We then discuss different types of in vitro models and fundamental elements that should be considered when developing an in vitro disease model. Finally, we review the current state and breakthroughs in the field of pancreatic in vitro models and conclude with some challenges that need to be addressed in the future development of in vitro models.

## 1. Introduction—The Need for In Vitro Models of the Pancreas

The fundamental goal of biomedical research is to decipher the molecular mechanisms of human disease in order to develop more effective or even new models for diagnosis, prevention, and therapeutic approaches. Due to ethical and other concerns, basic research on human (patho)physiology requires in vitro approaches. Current experiments generally rely on cell cultures and animal models (e.g., transgenic mice). While these are useful for certain aspects of disease modelling because they replicate disease phenotypes similar those observed in humans, it is becoming increasingly clear that basic genetic and molecular mechanisms may differ greatly between humans and other animals due to species-specific differences. For example, a recent systematic study of 5554 human genes and 4918 murine analogues showed that, in acute inflammatory stress responses changes in murine gene expression correlate poorly with those observed in humans [1]. Moreover, it is difficult to identify and independently vary the crucial molecular and cellular factors that contribute to disease in whole-animal models. This is probably the reason that many drugs fail to show efficacy and safety when translated from animal studies to human clinical trials [2].

The inadequacies of animal models to adequately recapitulate human disease, coupled with the associated ethical constraints and requirements to study disease pathogenesis under controlled conditions, have given rise to a new field at the interface of tissue engineering and pathophysiology, focused on the development of in vitro models of disease. These are defined as synthetic alternative experimental systems that contain living human cells and mimic tissue- and organ-level pathophysiology in vitro by taking advantage of recent advances in tissue engineering, microfabrication, and microfluidics [3,4,5]. When developing an in vitro disease model, some of the basic elements to consider are the source and type of cells, chemical and physical stimuli that promote the simulated cell phenotype, and additionally, when developing a 3D in vitro model, scaffold structure, fabrication process, and building blocks. The micro- and macro-environmental conditions of the in vitro models should be designed to mimic the characteristics of native or pathologically altered tissues. These include nutrient and metabolite concentrations, pH, and gas and mechanical gradients [6,7]. Due to the complex nature of human diseases and current limitations in systemic control, the full recapitulation of native pathophysiology in vitro remains a challenge. Even in the same organ, different responses may manifest depending on the influences of the physical and chemical microenvironment, immune and inflammatory responses, location of the disease, and whether the condition is acute or chronic [8]. By integrating multiple in vitro models into microfluidic circuits, a higher level of complexity can be achieved, which simulates the native environment to an even greater degree [4,9].

In vitro tissue and disease models serve as bioartificial surrogates for tissues and organs that can be used in basic research to investigate fundamental physiological phenomena, the molecular clues, and the processes involved in disease development and progression [10]. Moreover, such systems contribute to research that conforms to the “3R” (Replacement, Reduction, Refinement) guiding principles for more ethical research, as they can not only overcome the aforementioned limitations of animal models, but also reduce the need for in vivo testing [7].

In the case of the pancreas, particularly the islets of Langerhans, basic research into the development and evolution of diabetes largely relies on a variety of animal models (mainly rodents, particularly mice) at present [11]. Despite their many similarities, there are many structural and physiological differences between humans and rodents in the islets of Langerhans, leading to differences in functional coupling between cells and, ultimately, differences in the complex dynamics of insulin secretion [12]. This means that not all aspects of the results obtained in animal models can always be reliably extrapolated to humans. Moreover, current two-dimensional (2D) pancreatic culture models are unable to mimic the dynamics of insulin secretion in response to glucose or maintain beta cell viability over extended periods of time. In view of this, there is a great need for the development of increasingly sophisticated human in vitro models of the pancreas that more closely approximate pathophysiology at the tissue and organ levels, so that they recapitulate features that are essential to disease etiology and progression.

The research focusing on pancreatic in vitro models has already been described and assembled in some excellent reviews, especially those focusing on pancreatic cancer [13,14,15,16,17] and pancreatitis [18]. In view of this, this review provides an overview of in vitro tissue and disease models with a focus on the endocrine pancreas. Firstly, an overview of the anatomy, physiology, and pathophysiology of the human pancreas is provided, focusing on the islets of Langerhans and beta cell dysfunction. The basic elements that should be considered when developing an in vitro disease model, such as the type and source of cells, are discussed. This is followed by an overview of the current types of in vitro models, ranging from simple monolayer monoculture and co-culture models to self-assembled organoids and complex scaffold-based 3D microphysiological systems. As an integral part of 3D in vitro models, the importance of material selection and 3D bioprinting as an emerging technology for fabricating biomimetic scaffolds is addressed. Finally, breakthroughs in the field of in vitro models, focusing on bioreactors and organs-on-a-chip, are reviewed, and a future outlook is provided.

## 2. The Pancreas

### 2.1. Anatomy and Physiology of the Pancreas

Structurally, as well as functionally, the pancreas consists of two compartments: (i) the endocrine part, consisting of islets of Langerhans, and (ii) the exocrine part, with ductal and acinar cells. The latter accounts for 96–99% of the total pancreatic volume [12], and a schematic representation of the pancreatic anatomy is shown in Figure 1. The functional unit of the exocrine pancreas consist of an acinus and its corresponding draining ductile [19]. Pyramid-shaped acinar cells synthesize, store, and secrete digestive enzymes that are concentrated in a bicarbonate-rich fluid produced by ductal cells [20,21,22]. The secretions traverse through the ductal tree and drain into the intestinal lumen via the hepatopancreatic ampulla of Vater, allowing for the digestion of proteins, lipids, and carbohydrates [23]. The secretory process is mediated by cholinergic nerves that release acetylcholine and the peptic hormone cholecystokinin, which affects acinar cells, as well as secretin, which controls ductal cells [24,25,26]. The remaining 1–4% of the pancreatic volume belongs to the endocrine parenchyma in the form of endocrine micro-organs called islets of Langerhans. While the number of islets of Langerhans varies in different species depending on body mass and has been estimated to be between 3 and 15 million in the human pancreas [27,28], the size of Langerhans islets is constant in all species, with a diameter of 500–700 µm being the upper limit [29]. Each islet contains about a thousand endocrine cells of at least five different types. The most abundant are the insulin-secreting beta cells, which account for 50–60% of islet cells in humans and 60–80% in mice. These are followed by glucagon-secreting alpha cells (30–40% in humans, 15–20% in mice) and somatostatin-releasing delta cells (10–15% in humans and ~5% in mice). Less represented are PP or gamma cells, which secrete pancreatic polypeptide and epsilon cells, which secrete ghrelin [12,30]. Structurally, mouse islets have a core of beta cells surrounded by a mantle of non-beta cells, whereas in human islets the islet architecture appears to be more complex and diverse, promoting heterologous contacts between different cells [29]. Homotypic coupling of beta cells, heterotypic cell-to-cell contacts, and interactions with surrounding tissues play a crucial role in proper hormone secretion [31,32,33]. In contrast to the exocrine part, islet cells secrete their protein into the portal vein to regulate energy homeostasis. Beta cells play a central role in this process, with each cell containing ~10 000 insulin-filled secretory granules that are released upon stimulation by exocytosis [34]. Cells respond to glucose by closing ATP-sensitive K^+^ channels, membrane depolarization, Ca^2+^ influx through voltage-gated Ca^2+^ channels, and insulin secretion. While glucose is the primary secretagogue that activates the stimulus-secretion cascade in beta cells, several other nutrients (e.g., fatty acids, amino acids), hormones (e.g., glucagon-like peptide 1, glucagon, glucose-dependent insulinotropic peptide) and neurotransmitters (e.g., acetylcholine) potentiate or inhibit (e.g., somatostatin, leptin, epinephrine) glucose-induced insulin secretion, fine-tuning insulin secretion and subsequently maintaining glucose concentrations within a narrow range [12]. Since both hypo- and hyperglycaemia can have adverse health consequences, it is critical that beta cell function is precisely regulated [35,36,37].

### 2.2. Pathophysiology of Type 1 and Type 2 Diabetes Mellitus

Type 1 and type 2 diabetes mellitus are among the most common and important disorders of the function of the endocrine portion of the pancreas. Both can lead to acute and chronic complications, including cardiovascular problems, renal failure, neurological damage, vision loss, and overall increased patient mortality [38,39].

Type 1 diabetes (T1D) is a chronic, (auto)immune mediated disease caused by absolute insulin deficiency due to beta cell destruction [40]. The appearance of autoantibodies against insulin (IAA), glutamic acid decarboxylase (GADA), islet antigen-2 (IA-2 antigen) or Zn transporter are the first notable signs of immune dysregulation. While autoantibodies are an important biomarker of the disease, their role in beta cell loss is unclear [41]. The infiltration of immune cells around and within islets triggers and accelerates the development of T1D as islet antigens are exposed to the immune system [42,43]. It is not fully understood how the immune process is triggered, but a clear role of genetic predisposition associated with specific HLA alleles, particularly DR3-DQ2 and DR4-DQ8, has been established [44,45]. On the other hand, an increase in incidence in genetically stable populations in recent decades suggests that environmental factors are also necessary for disease development in a genetically susceptible individual [41,42]. However, recent evidence suggests that beta cells are not “victims” of autoimmune attack but are major contributors to the disease and that beta cell dysfunction precedes their destruction [43,46].

Type 2 diabetes is characterised by a progressive loss of insulin secretion by beta cells, frequently against the background of insulin resistance [47]. It is often described as a disease related to the modern, “Westernised” lifestyle, but is a complex polygenic disorder that is also thought to be influenced by a number of epigenetic and environmental factors (e.g., high caloric intake and lack of physical activity leading to obesity). The disease is a result of the complex interaction of insulin resistance and beta cell dysfunction. Initially, beta cells increase insulin secretion to compensate for peripheral tissue insulin resistance and maintain euglycemia. This compensation is possible through metabolic adaptation, which increases both the secretory capacity of beta cells and their cumulative mass [48,49]. However, chronic exposure to excessive fuel intake, especially hyperglycaemia, has deleterious effects on beta cells, leading to their dysfunction and insufficient insulin secretion and eventually to their failure, loss, and insulin deficiency [46,50,51]. With a reduction in beta cell mass and impaired insulin secretion by the remaining beta cells, hyperglycaemia worsens and completes the vicious cycle that ends in a complete cessation of insulin secretion [52]. Surprisingly, recent research suggests that beta cell function could be restored with rigid caloric restriction, challenging the assumed inevitably progressive nature of the disease and its irreversibility [47,53].

## 3. Cell Sources for Pancreatic In Vitro Models

### 3.1. Transformed Cell Lines

Most current in vitro modelling approaches depend on established, often transformed or immortalized cell lines that are prevented from reaching senescence, usually by one of the following mechanisms: deletion and/or mutation of senescence genes, overexpression or mutation of one or more oncogenes that override the action of senescence genes, or by blockage or loss of telomere function.

Such cells are cultured for many generations, resulting in significant changes in their genetic, epigenetic, and physiological properties, making them less suitable as models for native tissue cells [54]. Moreover, long-term cultivation leads to genetic adaptation, which contributes to heterogeneity in cultures of the same cell. The sum of genetic and epigenetic irregularities associated with long-term cultivation makes these cells less suitable for basic, functional, and pharmacogenomic studies [6].

The most commonly used insulin-secreting cell lines for in vitro modelling applications are insulinoma cells such as RIN, MIN or INS-1, hamster pancreatic beta cells (HIT), and beta-tumor cells (βTC). These cells produce insulin, as well as small amounts of glucagon and somatostatin. Some of them respond poorly to glucose; others respond well to glucose, but their concentration–dependence curve is clearly shifted to a higher sensitivity, compared with cells in native tissue sections. A comprehensive review of pancreatic beta cell lines, as well as their creation, physiology and application in diabetes mellitus research, has already been compiled in an excellent paper by Skelin et al. [55].

### 3.2. Primary Cell Lines

Modelling human disease in vitro depends primarily on the availability of tissue- and organ-specific cell types that accurately represent the modelled disease phenotype. Primary cells, derived from patients with or without the desired disease, represent the “gold standard” for in vitro disease modelling [56,57]. Primary cells are more representative of the functional unit of a tissue because they are isolated from tissue biopsies obtained from healthy or diseased patients. However, their use is not without hurdles. For example, they are difficult to obtain and isolate, are usually poorly characterized, often have a low proliferation rate, and have a relatively limited lifespan [58]. Another important aspect that should be carefully considered for primary cells is the number and density of cells so that the model truly represents the native functional unit of the replicated tissue [10].

The culturing and long-term expansion of isolated islets and primary islet cells has proven to be particularly difficult, limiting their use in in vitro studies over extended periods of time. The first problem with culturing arises from the demanding process of isolating and purifying viable pancreatic islets and cells within the islet. Pancreatic islets or primary beta cells (depending on the desired outcome of isolation) are usually obtained by mechanical disruption of the tissue, followed by enzymatic degradation. The most commonly used enzymes are trypsin, collagenase, elastase, hyaluronidase, pronase, DNase or dispase, alone or in various combinations [59]. However, shortly after isolation, Langerhans islet cells lose homo- and heterotypic intercellular contacts and are no longer capable of auto- and paracrine communication [55]. In addition, they lose the critical extracellular matrix (ECM) and basement membrane contacts. Interestingly, incompletely isolated islets that retain part of their native ECM exhibit significantly lower apoptosis rates and significantly better function in terms of insulin response in vitro compared to fully isolated and purified islets [60]. This implies that isolated islets rapidly lose mass, viability and function. The additional high metabolic demand and size of islets limit the availability and access of oxygen and nutrients to cells within the islet. Komatsu et al. showed that isolated human islets even require hyperoxic conditions (pO_2_ 270–350 mmHg) to maintain islet volume, higher islet viability and metabolism [61]. Due to their restricted diffusion and lower oxygen tension, apoptosis and necrotic cell death set in rapidly, especially in the islet core [62]. Beta cells are particularly affected and show reduced viability and loss of insulin secretion ability soon after isolation and cultivation on 2D flat substrate [63,64,65]. All this limits 2D in vitro pancreatic cell models to mimic the critical dynamics of insulin secretion upon stimulation with glucose [66]. Interestingly, culturing isolated islets in a microfluidic media flow results in higher cell density and better connections with surrounding cells (e.g., endothelial cells (ECs)) compared to islets cultured in conventional static 2D cultures. This positive effect on islet survival is explained by improved intercellular flow, which enhances albumin access to the centre of the islets. However, a high media flux can lead to attenuated glucose-stimulated metabolic and Ca^2+^ responses of beta cells at the periphery of islets, consistent with shear-induced damage [67].

### 3.3. Stem Cells

The limitations of primary cultures can be partially overcome by the use of stem cells, particularly induced pluripotent stem cells (iPSCs). Undifferentiated stem cells can be isolated from various sources (e.g., embryos, fetuses, adult tissues) and are capable of self-renewal and differentiation into multiple types of mature cells. Several attempts have been made to use iPSC-derived cells to model disease in vitro and elucidate disease mechanisms to aid in drug discovery and development. However, like other cell types, stem cell-based in vitro models have their own challenges: (i) controlling cell differentiation pathways (only) towards the desired cell phenotype and (ii) the presence of immature cell phenotypes. iPSCs allow for the in vitro modelling of a specific disease and the study of the mechanisms behind its development and progression, since they can be isolated from patients affected by a specific pathology. iPSCs are reprogrammed from somatic cells by ectopic overexpression of several transcription factors (e.g., Oct4, Sox2, Klf4 and c-Myc). They are self-renewing and can differentiate into various mature cell types [68].

Both hESCs and iPSCs have been shown to differentiate into pancreatic endocrine and exocrine progenitor cells via the activation or inhibition of key signalling pathways by growth factors and other chemical compounds such as WNT, activin, BMP, EGF, notch, etc. [69,70,71,72,73,74]. Moreover, the cells could be directed towards PDX1+ pancreatic endoderm, differentiating into monohormonal glucose-responsive beta cells under in vivo conditions [74,75]. However, the major drawback of hESCs/iPSCs for in vitro disease modelling is that ex vivo cultures exhibit largely immature phenotypes and polyhormonal cells preferentially become alpha rather than beta cells [76]. For example, in-vitro-derived insulin-expressing cells differentiated from human PSCs are polyhormonal and do not have the glucose response typical of mature beta cells [77]. Immature cells may be useful for studying the processes behind the early onset of disease, although it is unclear whether their biological responses can be extrapolated to mature cell types [78]. In this regard, a major advance was reported by Pagliuca et al. [79], who systematically tested >150 combinations of >70 compounds to formulate a 6-step protocol that generated ∼33% stem-cell-derived beta cells that resembled primary beta cells at the molecular, ultrastructural, and functional levels. These beta cells expressed specific canonical beta cell marker genes and possessed both developing and mature crystallized insulin granules in which normal insulin processing occurs. Functionally, these cells repeatedly increased intracellular Ca^2+^ and secreted insulin upon sequential glucose changes in vitro and rapidly restored euglycemia after transplantation in a diabetic mouse model [79]. Multiomics analyses identified three other important cell types, in addition to beta cells, among the final induction products, namely alpha-like cells, an unexpected population of enterochromaffin cells, and SOX9+ pancreatic progenitors that tend to give rise to exocrine cells upon further induction [80].

iPSC- and hESC-derived monohormonal glucose-responsive cells show tremendous promise for type 1DM research and the development of new therapies [74,81]. However, it is still unclear which combinations of physicochemical and mechanical signals are required to induce human iPSCs to commit to specific lineages. Robust methods for the directed differentiation and maturation of iPSCs into specialized and functional cell types need to be developed. This includes the use of combinatorial tissue engineering and microfabrication approaches, such as the use of biomimetic scaffolds, 3D bioprinting techniques, and microphysiological organ-on-a-chip systems that enable the combination of physicochemical and mechanical signals within the cell microenvironment for cell fate switching and the self-renewal of human PSCs [82].

### 3.4. Other Cell Sources

An alternative approach to proliferate human islets in vitro may also be nesidioblastosis, a process in which nonendocrine cells (e.g., ductal epithelium) differentiate into new pancreatic islets. As early as 1996, Kerr-Conte et al. demonstrated the formation of islet-associated cysts by culturing ductal epithelium within collagen-based matrices [83]. Later, other approaches succeeded in generating glucose-sensitive insulin-producing cells, e.g., by using umbilical cord-blood-derived mesenchymal stem cells [84] and human liver stem-like cells (HLSCs) [85]. A study by the Navarro–Tableros group developed a protocol using HLSCs to generate insulin-producing cells in vitro. The aggregation of HLSCs promoted spontaneous differentiation into cells expressing insulin and several key beta cell markers. In addition, the cells showed endocrine granules similar to those in human beta cells. In addition, they produced C-peptide after stimulation with high glucose. The cell constructs also significantly reduced hyperglycemia and restored a normo-glycemic profile when implanted into streptozotocin-diabetic immunodeficient mice [85].

A summary of various cell types used for in vitro disease models of the pancreas, showing their advantages and disadvantages, is shown in Figure 2.

## 4. Types of In Vitro Models of the Pancreas

There are several types of in vitro models that vary greatly in complexity, but all aim, albeit to varying degrees, to mimic certain (patho)physiological aspects of the target tissue or organ. We summarized different types of in vitro models based on their biological and/or structural complexity and briefly describe their advantages and disadvantages in Table 1. The listed types of in vitro models are described in more detail in the following chapters.

### 4.1. 2D pancreatic Cell Culture Models

The most widely used in vitro model to date is still the monoculture-based 2D monolayer, in which cells are isolated and plated on culture dishes or flasks and placed in cell culture incubators. Such simple 2D in vitro models are the more cost-effective and easy-to-handle testing platforms, especially compared to their complex 3D counterparts or animal models. They are particularly suitable for systematic and reproducible quantitative studies of cell physiology and drug development [86]. However, 2D cell cultures also have some disadvantages. For example, they cannot accurately represent the structure, function, and physiology of native tissues, so extrapolation of the information obtained from such in vitro studies to in vivo systems is often questionable. Culture dishes differ dramatically from native tissue, with the most obvious differences resulting from their stiff, smooth, flat, and chemically impermeable surfaces compared to the complex architectures of native tissue, as well as the liquid but relatively static environmental conditions that alter cell attachment, growth, and other responses [87]. The 2D environment of the culture dish polarizes the cells so that most of them are exposed to the culture medium from only one side. Since the cells only grow in 2D, only a fraction of the cellular surface is left for contact with neighbouring cells compared to native tissue. This leads to unnatural and irregular mechanostimulation that affects intracellular signalling, gene expression and, consequently, phenotypic fate [87]. Moreover, the concentrations of soluble factors that influence cell–cell communication, migration, and differentiation exhibit dynamic spatiotemporal gradients in vivo. In contrast, cells in 2D cultures experience homogeneous concentrations of nutrients, growth factors, and cytokines, resulting in atypical interactions with soluble factors. Ultimately, 2D substrates restrict cell growth to a planar environment and inhibit the formation of complex morphologies in vivo. Cells grown on 2D substrates generally exhibit flatter and more elongated morphologies. Interestingly, morphology alone influences gene expression and cellular processes such as proliferation, differentiation and apoptosis [88].

As with pancreatic cells, 2D substrates restrict growth, prevent the formation of complex morphology, and do not mimic the ECM cell-type contacts that are critical for normal endocrine cell function [89]. A recent advance in 2D monolayer islet cell cultures that promises to address some of the aforementioned shortcomings was described by Phelps et al., who coated glass surfaces with purified collagen IV or laminin in combination with a cell culture medium originally formulated for primary neurons. This supported the robust attachment and growth of primary human or rat islet cells as 2D monolayers. The islet cell monolayer cultures on glass stably maintained distinct monohormonal insulin+, glucagon+, somatostatin+ and PP+ cells and glucose-responsive synchronized Ca^2+^ signalling as well as the expression of transcription factors Pdx-1 and NKX-6.1 in beta cells [89]. This indicates that the ECM is one of the most important components of the islet cell microenvironment. Therefore, the focus should be on restoring the ECM environment to maintain islet cell function and viability. Further discussion of the restoration of the native ECM is part of the chapter 4.3. The Need for 3D In vitro Models. In brief, it has been shown that in vitro recapitulation of basic components (e.g., collagen I and IV [90], laminin [91], fibronectin and peptide (e.g., arginine-glycine-aspartic acid (RGD) sequence) epitopes [92]) of the peri-insular basement membrane of the pancreatic ECM, as well as synthetic biomimetic peptides [93], positively affect pancreatic islet viability and functionality even in 2D environments.

### 4.2. Pancreatic Co-Culture Models

Local and systemic cell–cell interactions regulate cell homeostasis and development and are an essential component of disease progression. In vitro models typically use a single cell type (monoculture), which reduces the number of possible interactions, whereas cells in vivo are in direct contact with various neighbouring cells while being simultaneously exposed to signalling molecules produced by distant cells and distributed throughout the body via endo-, exo-, or paracrine signalling.

To overcome the loss of islet quality during culture, researchers usually co-culture islets with mesenchymal stem cells (MSCs) [94]. In a systematic review, de Souza et al. performed a meta-analysis to investigate whether MSC-secreted factors are sufficient to improve islet quality or whether physical contact between MSCs and islets is required. Mean viability was higher in islets co-cultured with MSCs than in islets that were cultured alone. The improvement in viability was higher in islets co-cultured in indirect or mixed contact with MSCs than in direct physical contact. Moreover, the mean value of insulin stimulation index was higher in islets from co-culture than in islets cultured alone, regardless of the contact system. Overall, the results showed that co-culturing islets with MSCs has the potential to protect islets from injury during the culture period [94]. Jun et al. also suggested that adipose-derived stem cells (ADSCs) may have protective effects on isolated islets. Islets cocultured with ADSCs showed significantly different ultrastructural morphologies, higher viability, and improved insulin secretion compared with monocultured islets. Furthermore, in vivo experiments with xenotransplantation of microfiber-encapsulated islet spheroids into a mouse model of diabetes showed that co-cultured transplanted mice maintained blood glucose levels longer than mono-cultured transplanted mice and required less islet mass to reverse diabetes [95].

Recent studies have also described the role of blood vessels in promoting embryonic development of the pancreas, suggesting a crucial role of islet-associated endothelial cells (ECs) in islet survival, proliferation, and differentiation. As mentioned previously, islets are highly vascularized in vivo, while associated ECs facilitate blood glucose sensing and hormone secretion. Although the exact mechanism behind this is poorly understood, the coupling between ECs and beta cells appears to be critical for the proliferation and survival of both cell types in isolated islets. Coupling is maintained for a short time after isolation, but ECs in culture slowly begin to lose density and morphology and eventually deteriorate. To address this problem, Kaufman-Francis developed a tri-culture setup in which human ECs, foreskin fibroblasts, and mouse islets were seeded on porous poly(lactic-co-glycolic acid)/poly(L-lactic acid) (PLGA/PLA) scaffolds. They showed that islet survival and insulin secretion were improved compared to the culture without ECs. Moreover, the addition of foreskin fibroblasts to islet-endothelial cultures supported islet survival as well as insulin secretion. Gene expression of growth factors, ECM components and differentiation markers was significantly different in 3D versus 2D culture models and was further improved after the addition of fibroblasts [96]. Recently, Lazzari et al. adopted a scaffold-free approach and were the first to report a triple co-culture spheroid consisting of pancreatic cells, fibroblasts and ECs. The co-culture model successfully replicated fibrotic tissue and collapsed vascular structure, both hallmarks of pancreatic ductal adenocarcinoma. An evaluation of spheroid growth and treatment response confirmed the impact of the microenvironment on the drug sensitivity of pancreatic cancer cells and demonstrated the ability of this model to replicate the treatment resistance often observed in vivo [97]. Jun et al. also investigated the supportive relationships formed by cell-to-cell contacts between primary pancreatic islet cells and hepatocytes. Using concave microwells, they developed hybrid 3D coculture systems with different ratios of hepatocytes to islet cells and evaluated characteristic functions of the hybrid spheroids compared to those of islet or hepatocyte monoculture models. They found that the hybrid models formed spheroids with higher stability and tighter cell–cell junctions, exhibiting a higher viability over 7 days compared to the monoculture spheroids. In addition, they analysed the effect of hepatocytes on insulin release from islet cells in the hybrid spheroids and found that models containing less than 25% hepatocytes maintained glucose response at comparable levels to the islet mono-spheroids. The function of cocultured spheroids in controlling glucose was evaluated in vivo, which resulted in the rescue of diabetic mice with consistent maintenance of normal blood glucose levels over 4 weeks [98].

### 4.3. The Need for 3D In vitro Models

The examples described above highlight the advantages of 3D in vitro systems compared to traditional flat culture models. It is well documented that 3D cell cultures establish cell–cell and cell–ECM interactions that better mimic the biochemistry and mechanics of the native cell microenvironment in vivo [99]. As mentioned above, non-physiological cell culture conditions lead to long-term loss of function or even cell death. In a 3D in vitro model, cells grow and interact with each other and the ECM more effectively and in all spatial dimensions. Therefore, they are exposed to a variety of position-specific stimuli (e.g., physical forces and soluble factors) that actively influence cell function and gene expression profiles. These stimuli can generally be divided into three groups: (i) the physical and structural stimuli from the 3D microenvironment of the cell, (ii) the biochemical cues from coupled cells and the ECM, and (iii) the physicochemical signals emanating from various gradients (e.g., concentration, temperature, pH, gas gradients). The three groups of cues, which act on cells in a 3D in vitro model, are shown in Figure 3.

The first group, namely, the structural stimuli, are modelled in vitro by choosing the appropriate material, with the aim of replicating the surface and mechanical properties of the native microenvironment at multiple scales (e.g., nanoroughness microarchitecture, stiffness, elasticity). In vivo, cells are subjected to extracellular and intracellular mechanical forces that control their behaviour. By modifying the surrounding ECM (in the case of an in vitro model of the surrounding material), cells respond to dynamic mechanical signals, such as osmotic and hydrostatic pressure, stress, strain, fluid flow, and shear stress [100,101].

The second group includes the biochemistry of the microenvironment, which affects cell division, differentiation, morphology, migration, apoptosis, and the secretion of cells, as well as ECM components. To accurately model these in vitro, appropriate biochemical cues should be incorporated into the experimental system (in vitro model), through the use of 3D-structured biomaterials and appropriate cell orientation within and possibly with the addition of external factors (e.g., using microfluidics). A biomaterial, usually defined by its function, is a natural or synthetic material designed to interact with biological systems [102]. As such, they must be optimised for their purpose to have the correct mechanical, chemical, and biological properties. The latter usually involves the incorporation of living cells or various biochemical factors into the material. Two different strategies for the incorporation of biochemical factors (e.g., growth factors) into biomaterials have been pursued: (i) chemical conjugation to scaffold materials and (ii) physical encapsulation with optional controlled release. The former strategy generally involves chemical binding between cells and the soluble factor-containing substrate. The latter approach is achieved by the encapsulation, diffusion and pre-programmed release of soluble factors from the substrate [103,104].

Due to the consumption of nutrients and production of metabolites, chemical gradients emerge within the system that influence cellular behaviour depending on their position in the construct. Cells in the core are typically exposed to lower concentrations of nutrients and higher concentrations of metabolites compared to cells at the surface of the construct. A functional 3D in vitro model may be limited by the diffusion of nutrients and oxygen due to its thickness. Oxygen concentration is probably one of the most important physio-chemical cues. While the atmospheric oxygen tension is easily controlled in hypoxic incubators, the actual oxygen tension at the cell surface can vary greatly due to the low solubility of oxygen. It mainly depends on the height of the medium and the thickness of the biomaterial and the final 3D construct [105]. Coupling chemical and mechanical signals can prevent local concentration drops or increases and overcome diffusion limitations. Microfluidic devices and bioreactors, discussed in the following chapters, are designed to control gradients and reproduce in-vivo-like conditions. They provide mechanical and chemical stimuli, enable real-time monitoring, and offer the possibility of subtle spatiotemporal modulation of culture conditions that favour the desired cell phenotype [10,104].

Further supporting the previously unmentioned importance of the 3D environment specifically for the differentiation, growth and functionality of islets of Langerhans is the fact that a direct correlation between pancreatic islet morphology and endocrine differentiation has recently been confirmed. During endocrine progenitor cell differentiation, the developing alpha cells first migrate to the outside of the islet and form a mantle, whereas the later-forming beta cells remain in the core of the islet. This spatiotemporal proportionality leads to the typical spherical 3D architecture of the human pancreatic islet (alpha cell mantle and beta cell core), which is essential for normal islet function [106]. Therefore, to accurately study cell and tissue (pato)physiology in vitro, pancreatic cells should be grown in 3D microenvironments that recapitulate the necessary biochemical and mechanical signals present in native tissue ECM while allowing for hierarchical processes such as tissue organization and cell migration. Cells can be assembled in a 3D construct by using support-free organoid cell cultures or by embedding or seeding cells onto a 3D matrix support (scaffolds) [107].

#### 4.3.1. Organoids

Organoids, often incorrectly referred to as spheroids, are 3D cellular constructs ex vivo, usually derived from PSCs or organ-restricted adult stem cells (ASCs) that self-assemble or are directed to self-assemble under certain organogenesis cues [108]. They physically and physiologically recapitulate the architecture, composition, cellular organization, and functions of an organ. In contrast, spheroids are simple cell aggregates that have little to no relevant tissue structure. Organoids are constructed based on secreted soluble and both modular and non-modular ECM signals. To date, research on organoids has mostly focused on PSCs.

The importance of the 3D microenvironment in vitro islet development from PSCs was ignored in early attempts to establish islet-like cells in vitro, which was partly responsible for the immaturity of the derived cells due to reduced growth factor gradients. In contrast, when PSCs are grown in a 3D microenvironment, they self-organize, develop physiologically relevant cellular patterns, and eventually differentiate into various endoderm- and ectoderm-derived tissues [109]. Direct evidence for the influence of 3D structure on islet organoid maturation was provided by comparing 2D stem-cell-derived beta cells and their clustered counterparts. The static suspension culture of dissociated beta cells generated ∼50–150 μm sized endocrine cell clusters with significantly increased co-expression of PDX1/NKX6.1 and PDX1/GLUT1, reduced expression of MAFB, a marker of endocrine progenitor cells, and enhanced functional maturation [110]. The ability of hPSCs to self-assemble under 3D conditions (usually using orbital shakers, non-adherent tissue culture plates or the hanging drop method) has been widely used to aggregate dissociated hPSCs into 3D organoids with variable diameters depending on the culture system. For example, Pagliuca et al. used a stirring plate to constantly rotate hPSCs cultured in spinner flasks to generate ∼100–200 μm-sized cell clusters. Morphologically and functionally, these clusters resembled native human islets; however, few non-beta endocrine cells were detected compared to those in human islets [79,111]. Guo-Parke also showed that islet models prepared by the self-assembly method exhibited a 1.7- to 12.5-fold increase in insulin secretion in response to acute stimulation (e.g., glucose, amino acids, incretin hormones, or drugs) compared with equivalent cell monolayers. Moreover, glucose-stimulated insulin secretion was increased 6-fold in islet models, whereas it was increased only 3-fold in monolayer cultures when glucose concentration was increased from 2 to 20 mmol/L [112].

To better control the self-assembly process of 3D structures, other strategies used ECM components as scaffolds to promote 3D structure formation and cell–matrix interactions. Greggio et al. established 3D culture conditions in a Matrigel-based system that allowed the culturing and expansion of mouse embryonic pancreatic progenitor cells in vitro. By manipulating the media composition, they generated either hollow spheres, composed mainly of pancreatic progenitor cells, or complex organoids that spontaneously underwent pancreatic morphogenesis and differentiation. The system recapitulated pancreatic development and showed both exocrine (acinar) and endocrine (insulin+) cell differentiation in vitro. It is important to note that although the system allowed for the study of pancreatic development in vitro, it did not support the long-term expansion of pancreatic cells [113]. Moreover, the organoid structures mentioned above consisted exclusively of ductal cells expressing the embryonic progenitor marker Pdx1, and the differentiation of these cells into endocrine lineages in vitro remained elusive. In contrast, Hutch et al. set conditions in a 3D Matrigel system to induce the Wnt-Lgr5-Rspo axis and showed that adult pancreatic organoid cultures could undergo long-term expansion in vitro in the presence of FGF10, Nog, Rspo1 and EGF. Furthermore, when the researchers incorporated adult duct-derived organoid cells into embryonic rat pancreas and transplanted this mixture under the kidney capsule of immunodeficient mice, most cells differentiated into three mature pancreatic lineages: ductal, acinar, and endocrine. Among the cells in the endocrine lineage, most were fully differentiated monohormonal cells expressing only glucagon or somatostatin. Cells expressing only insulin were differentiated with much lower efficiency [114]. Recently, Candiello et al. developed a novel platform “Amikagel” by polymerizing amikacin hydrate and polyethyleneglycol diglycidyl ether (PEGDE), which facilitates the self-aggregation of hESC-derived pancreatic progenitor cells and their coaggregation with supporting endothelial cells. Amikagel-produced 3D cell aggregates (Figure 4) had increased insulin/C-peptide expression and enhanced glucose responsiveness [115]. In addition, islet organoids cultured in a microporous scaffold showed improved control over islet organoid size and cell–cell interactions. These ∼250–425 μm sized islet organoids showed more mature marker expression and performed better in GSIS than their suspension culture counterparts [116]. One method that promises to resolve the variability in the size of self-assembled organoids is microcontact printing. This technique allows for the manipulation of aggregate shape and size by changing the size of the printed area. For example, this technique was used to seed stem cells into circular patches of 120 mm in diameter using the covalent microcontact printing of laminin adhered to glass coverslips. The stem cells then aggregated into clusters and were further differentiated into pancreatic endocrine precursors. Van Hoof et al. showed that microcontact printing provides an efficient way to produce the uniformly clustered functional beta cells for diabetes research [117].

Due to their endocrine function, islets are among the most vascularized organs. In this regard, vascularization is critical for islet development and endocrine function, and a lack of vascular networks not only reduces the fidelity of islet organoids but also their viability during in vitro culture. Taniguchi’s group attempted to achieve in vitro vascularization by coculturing ECs with endocrine cells. They reported that the coculture of cell lines, native tissue fragments, and iPSC spheroids with human umbilical vein endothelial cells (HUVECs) and MSCs in Matrigel allowed for the formation of vascularized islet organoids Furthermore, the gene expression patterns of the vascularized islet organoids better reflected the native islets than those of non-vascularized islets [118,119]. Additionally, the use of microfluidic devices could drastically improve the culturing of organoids and spheroids by allowing them to grow and function under conditions similar to those in vivo [120].

As the developmental and functional characteristics of islet organoids have become increasingly representative of physiological human islets, these “organs in a dish” have been used to understand pancreatic stem cell biology, organogenesis, and human pathologies. As current in vitro model systems limit the study of diabetes and other disease mechanisms, islet organoids are creating new opportunities to study diabetes. Currently, iPSCs derived from patients with type 1 DM are being used to generate new models demonstrating that the cells respond in vitro to various forms of beta cell stress [81]. An analysis of three type 2 DM-related genes, CDKAL1, KCNQ1, and KCNJ11, in gene-edited hESCs revealed normal endocrine specification but impaired insulin secretion in vitro and in vivo [121]. Wolfram syndrome, an autosomal recessive disease caused by mutations in WFS1, is characterized by juvenile diabetes. Beta-like cells differentiated from iPSCs of Wolfram syndrome patients showed decreased insulin content and increased activity of molecules related to endoplasmic reticulum stress, suggesting that beta cell failure is caused by WFS1 deficiency [122]. For a more detailed overview of disease modelling with organoids, we refer the reader to an article by Clevers, who published a comprehensive review of “Disease modelling with organoids” [123], as well as a recent paper by Zhang et al. that specifically focuses on the review of organoids as “Promising in vitro models for diabetes” [124].

#### 4.3.2. In vitro Modelling with 3D Biomimetic Scaffolds

In vivo, pancreatic islets grow embedded in the ECM, a non-cellular component of tissues composed of numerous proteins (e.g., collagens, laminins, fibronectins), glycoproteins, glycosaminoglycans, and growth factors that interact to form complex 3D structures that provide cells with biomechanical support and receptor-mediated intracellular signals essential for tissue development and homeostasis. Cells sense these mechanical signals through interactions between cell surface integrins and ECM proteins. Hydrated proteoglycans fill the interstitial cavities and sequester soluble biomolecules such as growth factors, cytokines and signalling molecules. Cells dynamically restructure the microenvironment to release signalling molecules, enable migration, or allow cell function and the deposition of ECM components [125]. Furthermore, in their ground-breaking work, Petersen et al. also showed that phenotype can override genotype solely through interactions with the ECM, highlighting that the ECM contributes to complex spatiotemporal interactions that control cell phenotype [126].

Although the architecture and physical interactions within and around islets are complex and not fully understood, there is strong evidence that islets, like other tissues, are strongly influenced by cell–ECM interactions. In mature, intact islets, interactions with the ECM or synthetic matrix materials have been shown to regulate survival, insulin secretion, and proliferation, and to contribute to the maintenance and restoration of spherical islet morphology. Furthermore, beta cell–ECM interactions also play a critical role in the activation of the NF-κB signalling pathway, a critical pro-inflammatory regulator [127]. Similarly, ECM-based materials have been shown to regulate survival, proliferation and insulin secretion in purified beta cell cultures [128]. Recently Li et al. demonstrated the importance of cell–cell and cell–ECM interactions for beta cell survival, viability, and function, in a study in which they established an in vitro cell microenvironment by culturing beta cells in contact with soft polymer microbeads that served as “synthetic neighbours.” The microbeads were additionally modified with cell–cell signalling factors (cell surface receptor and its membrane-bound ligand pair EphA/EphrinA) and coated with tissue-specific ECM building blocks. This densely packed biomimetic in vitro microenvironment was able to promote both native cell–cell and cell–ECM interactions while enhancing beta cell viability and supporting insulin production for up to 3 weeks [129].

Scaffold surface topography, chemistry (e.g., wettability, stiffness, roughness), microstructure (e.g., porosity, pore size, pore shape), and mechanical properties significantly influence cell behaviour [130,131]. In the case of pancreatic cells, it has been shown that cells cultured on 3D scaffolds are able to differentiate into a physiologically relevant tissue in vitro and that the morphology is very different from that of cells grown on 2D substrates. It has been shown that 3D scaffolds promote greater cell aggregation, proliferation, differentiation, long-term survival and improved function of pancreatic cells compared to 2D substrates [132].

#### 4.3.3. Pancreatic Tissue-Derived Scaffolds

The strategy to develop biomimetic scaffolds for specific (pato) physiological functions requires a comprehensive understanding of the tissue composition and the localization of the bioactive components of the ECM. Moreover, the selection of an appropriate material for scaffold fabrication should take into account that pathological tissues exhibit altered ECM properties compared to their healthy counterparts [125]. Interestingly, during the pathogenesis of type 1 DM, certain components of islet ECM that form basement membranes and the interstitial matrix of islets and, surprisingly, the intracellular composition of islet beta cells themselves, are significantly altered [133]. Accordingly, when developing an in vitro disease model, the scaffold should recapitulate the altered ECM properties. It should be emphasized that further studies are needed to understand how the mechanical properties of pancreatic tissue change under pathological conditions, such as chronic pancreatitis [134]. Although synthetic and natural materials provide important biochemical cues, existing scaffolds are unable to fully emulate the complexity of the healthy or pathologically altered pancreatic ECM. Tissue decellularization methods could provide intact ECM scaffolds (decellularized extracellular matrices (dECM)) for the development of more complex in vitro models [135]. dECM refers to the extracellular matrices obtained from different parts of tissues and/or organs through a decellularization process. Pancreatic dECM can be obtained by various decellularization protocols, but the protocol generally involves lysis and removal of the cellular components of the tissue by perfusion with salt and detergent solutions. This leaves behind the tissue-specific ECM. While the composition and material properties of dECM may vary depending on the tissue source and processing method, the composition generally consists of common proteins/macromolecules such as collagen, laminin, fibronectin, elastin and their tissue-specific glycosaminoglycans, cytokines and growth factors [136].

The research team led by Goh et al. was one of the first to successfully decellularize the entire pancreas, demonstrating that the pancreatic ECM can be used as a scaffold to support and improve pancreatic cell functionality. Using a perfusion-based decellularization process, they effectively removed cellular and nuclear material while retaining a complex 3D microarchitecture with perfusable vasculature and a ductal network, as well as key ECM components. To mimic pancreatic cell composition, they recellularized the entire pancreatic scaffold with acinar and beta cell lines and cultured it for up to 5 days. Their result showed successful cellular engraftment within the decellularized pancreas, and the resulting graft strongly upregulated insulin expression [137]. Recently, Jiang et al. optimized the decellularization protocols and developed pancreatic dECM hydrogels with different viscoelastic properties that correlated with matrix composition. In situ 3D encapsulation of human or rat islets in dECM hydrogels resulted in preserved glucose-stimulated insulin release. The composition and morphology of the islets was also altered, with an increased retention of endothelial cells located in the islets and the formation of cord-like structures or sprouts arising from the islet spheroid. These multicellular miniorgans also showed dynamic interactions with the dECM hydrogel, forming protrusions during culture that consisted of fibroblast-like sprouts and served as channels for endothelial cell migration [138].

It should be noted that even the best dECMs are unable to perfectly mimic the complex ECM, as the highly specific spatial positioning of each structural protein within the native tissue is disrupted during the process [139]. One of the challenges is to find a balance between the complete removal of cellular components and the preservation of fine vessels and other tissue structures. In addition, some toxicity has been observed when cells are grown on decellularized tissue scaffolds, possibly due to the retention of detergents from decellularization within the ECM [140]. dECM scaffolds are neither sustainable nor without ethical concerns, making them less suitable for scalable production. However, in terms of composition and structural properties, they remain the closest approximation to ideal scaffolds and can serve as a comparison when designing new ECM substitutes.

#### 4.3.4. Biomimetic Scaffolds from Synthetic Polymers

Synthetic polymeric scaffolds have been extensively studied for islet culturing due to their stability, reproducibility, and ease of functionalization. Mao and colleagues constructed PLGA scaffolds with pore sizes of 100 and 300 μm, seeded them with islet-like cells, and then transplanted them into diabetic immunodeficient mice, resulting in normoglycemia [141]. Similarly, Gibly et al. demonstrated the ability of a macroporous PLGA scaffold to enhance islet survival and engraftment in both rodent and porcine models while allowing for cellular infiltration and revascularization [142]. Blomeier et al. fabricated microporous polymer scaffolds from copolymers of lactide and glycolide and transplanted them into mice. The islets grown on the scaffolds remained localized at the graft site and survived for extended periods of time. Moreover, they retained their native architecture and developed functional vasculature [143]. In another study, Chun et al. grew cells isolated from cells of pancreatic islets of Wistar rats on fibrous polyglycolide (PGA) scaffolds coated with poly-l-lysine. The scaffolds promoted cell adhesion and the cells cultured on PGA scaffolds showed native-like morphology, improved viability and increased insulin secretion [144].

Synthetic highly porous scaffolds also offer the possibility of culturing higher cell densities required to mimic tissues in vitro. Insufficient oxygen tension is one of the main obstacles to the success of cell-based constructs, especially when they consist of cells with high metabolic rates, such as pancreatic islets. Indeed, hypoxia negatively affects islet cells survival, mainly through the stabilization of hypoxia-inducible factor (HIF)-1α and the subsequent activation of its target genes, leading to a cascade of events that ends in islet cell apoptosis. Moreover, HIF-1α has been shown to lead to impaired glucose responsiveness [145]. Two promising approaches to address this challenge in vitro are the production of highly porous scaffolds and by revascularisation guidance. For example, islet viability was improved in microporous poly(dimethylsiloxane) (PDMS) scaffolds under low-oxygen culture conditions compared to microporous scaffolds of comparable size. Moreover, the stabilization of glycemia and independence from exogenous insulin was observed in rodent models [146]. Recently, Tokito et al. also used synthetic oxygen-permeable PDMS scaffolds to achieve high native-like seeding densities and culture islet-like tissue in vitro. The cultivation of 3D islet-like tissue from a rat pancreatic beta cell line on the scaffolds with external oxygenation resulted in densities and functions (assessed by proliferation and insulin secretion) that were more than threefold, and almost twofold, higher, respectively, than without oxygenation [147].

To recapitulate the native microenvironment of islets, it is useful to further functionalize synthetic polymer scaffolds with bioactive molecules (e.g., ECM building blocks and growth factors). For example, Weber et al. incorporated ECM components into poly(ethylene glycol)-based scaffolds (PEG), resulting in a reduction in apoptotic signalling and enhanced glucose-stimulated insulin- secretion, for both a beta cell line and murine islets [148]. Translating this in vivo, Salvay et al. fabricated microporous PLGA scaffolds and functionalized them with collagen IV, fibronectin, laminin-332 or serum proteins before seeding them with murine islets and implanting them in mice. They showed that ECM proteins adsorbed to microporous synthetic scaffolds improved islet function, while collagen IV maximized graft function compared to the other proteins tested [149]. Another approach investigated peptide amphiphiles containing RGD in addition to a matrix metalloproteinase-2 (MMP-2)-sensitive sequence for beta cell encapsulation. Peptide hydrogels resulted in improved retention of glucose response and islet morphology in culture [150]. Recently, Marchioli et al. investigated the possibility of functionalizing synthetic scaffolds with vascular endothelial growth factor (VEGF) to promote the *de novo* vessel formation. The scaffolds consisted of polycaprolactone (PCL) in the form of a 3D-shaped ring with a heparinized surface for electrostatic binding of VEGF. Human islets of Langerhans were contained in an alginate core in the centre of the macroporous ring. They showed that a relatively small amount of VEGF could efficiently induce neovascularization around and through the pores of the scaffold. Heparin covalently bound to the surface of the scaffold allowed for the efficient sequestration of VEGF, while the islets encapsulated in the alginate core showed a functional response to glucose comparable to free floating islets even after 7 days [151]. A similar approach was followed by the group led by Samuel Stupp, who used heparin-binding peptide amphiphiles doped with VEGF and fibroblast growth factor (FGF) to encapsulate islets. Their incorporation resulted in an increase in islet endothelial cell sprouting, improved viability and glucose stimulation indices, and increased efficacy of islet transplantation in a syngeneic mouse model [152].

#### 4.3.5. Biomimetic Scaffolds from Naturally Derived Polymers

The synthetic polymers such as PGA and PLGA, although successful to some extent in establishing pancreas in vitro models, are typically very stiff (with Young’s modulus in the range of several GPa [107]) compared to a native pancreas (with a nonlinear shear stiffness of 1–2 kPa [153]). In terms of their mechanics, hydrogels are closest to the required range, and can be made with many naturally derived (e.g., alginate, gelatine, cellulose) polymers [154]. Hydrogel scaffolds also have other advantages for beta cell survival, allowing for the easy diffusion of hormones, metabolites, and nutrients with low molecular weight. In addition, they are affordable, widely accessible, and highly biocompatible [154]. Moreover, to promote the desired cell responses and phenotype, they offer the possibility to further (bio)functionalize the material surface with bioactive molecules (e.g., ECM components, growth factors, adhesion factors) without altering the structural and mechanical properties, which is more difficult to achieve with their synthetic counterparts [155]. For example, Phelps et al. developed injectable hydrogels to improve the vascularization and engraftment of transplanted pancreatic islets. They additionally incorporated the proangiogenic factor VEGF-A and cell-adhesive peptides into proteolytically degradable hydrogels, which significantly improved islet vascularization and function and led to the complete reversal of hyperglycemia in diabetic mice [156]. In a recent study, Borg et al. (bio)functionalized macroporous heparine/PEG-based hydrogel scaffolds with adhesion-mediating peptide ligands—RGD (cyclo(Arg-Gly-Asp-D-Tyr-Lys)). The scaffolds were used to house pancreatic islets in the form of a 3D co-culture, with adherent MSCs serving as accessory cells. The architecture of the scaffolds with interconnected macropores allowed for the unimpeded exchange of nutrients and provided mechanical protection while maintaining the 3D distribution of MSCs and islets. MSCs adhered to adhesion peptides bound to the scaffold and enhanced the displayed artificial 3D niche by secreting ECM proteins, allowing for proper cell–matrix interaction. The islets survived seeding into the scaffolds and secreted insulin after glucose stimulation in vitro. The constructs remained intact and functional 7 days after subcutaneous transplantation in mice [157]. By developing islet organoids from human embryonic stem cells (hESCs) within collagen type 1 scaffolds, Wang et al. were among the first to combine organoid technology with biomimetic scaffolds. The organoids formed consisted of pancreatic alpha, beta, delta and gamma cells. The high co-expression of PDX1, NKX6.1 and NGN3 in these cells indicated the characteristics of pancreatic beta cells. More importantly, most insulin-secreting cells generated did not express glucagon, somatostatin, or pancreatic polypeptide. The expression of mature beta cell marker genes such as Pdx1, Ngn3, insulin, MafA and Glut2 was also detected and high expression of C-peptide confirmed *de novo* endogenous insulin production. Insulin-secretory granules, an indication of beta cell maturity, were also detected in these cells. These “mature-like” cells were also sensitive to glucose, with exposure of the cells to a high glucose concentration inducing a sharp increase in insulin secretion [158].

Although beyond the scope of this review, it is also worth noting that macro- and microencapsulation of islets of Langerhans in hydrogels has also shown promise as a therapy for type 1 DM. The hydrogel capsule is permeable to metabolites, hormones, and nutrients while preventing immune cells or larger molecules (e.g., antibodies) from coming into contact with the transplanted islets, ultimately prolonging islet survival and functionality [159].

##### 4.3.6. 3D Bioprinting of Biomimetic Scaffolds

Conventional methods for fabricating biomimetic scaffolds (e.g., hydrogel casting, chemical/gas foaming, fibre bonding, salt leaching, freeze-drying, electrospinning) provide control over the bulk properties of the scaffold, but are often insufficient to produce accurate and reproducible pore size, shape, network, internal architecture, and topology. Other limitations include suboptimal cell distribution due to inaccuracies associated with the cell seeding process. Pancreatic cells need to be precisely arranged according to the function of the tissue; thus, the above methods typically result in oversimplified 3D in vitro models [160]. In this regard, additive manufacturing could be a viable alternative for the process of living tissue fabrication. Through precise spatial deposition of cells, ECM components, growth factors, and other components, 3D bioprinting has the potential to produce sophisticated biomimetic scaffolds and tissue constructs. The many advantages of this technology include automation, precision, reproducibility, geometric freedom and control (macromorphology, pore size, porosity, interconnectivity), and compatibility with a wide range of materials (depending on the technique). This technology also enables the fabrication of intrinsic composition gradients and control over cell densities [161,162].

The first application of 3D bioprinting for in vitro modelling of the pancreas was reported by Daoud et al. [163], who seeded islets into synthetic poly(DL-lactide-co-glycolide) scaffolds. They also investigated the effects of different 3D environments on islets by incorporating ECM components such as collagen I, fibronectin and collagen IV. Their results showed that the procedure allowed the fabrication of highly uniform human islets within the scaffolds and that incorporation of ECM components into the scaffolds improved long-term culture and led to a similar insulin release profile to that of freshly isolated islets. Gene expression was significantly increased for all pancreatic genes, resulting in a ~50-fold increase in insulin gene expression compared to suspension culture. The distribution and presence of pancreatic hormones was also greatly increased [163]. The major drawback of this study was the discrepancy between the mechanical properties of the scaffold used compared to native pancreas. Marchioli et al. improved on this and used 3D bioprinting to fabricate a porous alginate-based scaffold (Figure 5) that served as an extrahepatic islet delivery system [164]. They successfully embedded INS1E beta cells and human and mouse islets into the bioprinted scaffolds without compromising their morphology and viability, while preventing their aggregation. However, the high viscosity of the material required for bioprinting resulted in impaired glucose diffusion and limited functionality of the islets. Interestingly, full functionality of the islets was restored after they were removed from the hydrogel [164]. More recently, Duin et al. developed a printable hydrogel formulation based on alginate and methylcellulose that did not impair the diffusion of relevant macromolecules such as glucose and insulin into the scaffold. After encapsulation in the hydrogel and bioprinting of macroporous constructs, the islets retained their viability, morphology and functionality. Furthermore, the localization of alpha and beta cells within the islets was preserved and the printed islets continued to respond to glucose stimulation and produce insulin and glucagon for up to 7 days in culture [165].

In this regard, our group has also contributed to the state of the art by developing 3D-printed polysaccharide-based biomimetic scaffolds (alginate, carboxymethylcellulose and nanofibrillated cellulose) with tailored properties [166]. With the intention of fine-tuning the 3D printability, rheological, mechanical, swelling, degradation and surface properties of the hydrogels, variable concentrations of NiCu nanoparticles were incorporated into the hydrogels. We demonstrated that NiCu nanoparticles were an effective tool for controlling hydrogel viscosity and scaffold swelling, degradation, and topographic properties. All scaffolds also promoted cell adhesion, cell aggregation, and cell migration, and supported the long-term growth of pancreatic cells, which also exhibited more physiologically relevant morphology [167]. A major step towards the development of biomimetic materials for the fabrication of pancreatic constructs was recently taken by Kim et al., who developed a pancreatic dECM bioink suitable for 3D bioprinting. Specifically, the insulin secretion and maturation of insulin-producing cells derived from human pluripotent stem cells were strongly upregulated when cultured in dECM bioink. Moreover, co-culture with human-umbilical-vein-derived endothelial cells reduced central islet necrosis under 3D culture conditions [168].

A disadvantage of extrusion-based bioprinting is the direct exposure of cells to shear stress. By examining how bioprinting affects islet viability, Klak et al. demonstrated that pancreatic islets are highly susceptible to shear stress. The viability of pancreatic islets and cells is closely related to the applied pressure and the resulting shear stress during printing, with higher values of these parameters leading to decreased cell viability [169]. While the impact of shear stress can be minimised by the careful fine-tuning of printing parameters (printing pressure, hydrogel viscosity, nozzle diameter), direct extrusion also has other limitations, such as the geometric complexity of scaffolds (e.g., hollow 3D networks) [170]. Nonetheless, extrusion-based techniques are highly adaptable and can be combined with more sophisticated nozzles or additional tools that allow for the printing of support baths [171,172], microfluidic and coaxial printing techniques [173,174,175] that allow the fabrication of scaffolds with integrated hollow channels, either by sacrificial ink deposition or by “core-shell” printing, where the main ink surrounds a crosslinking solution and forms a hollow tube during deposition. Liu et al. further improved the coaxial bioprinting strategy by combining the advantages of macro- and microencapsulation of islets. The approach involves the use of a specially designed coaxial printer and an alginate/gelatine-based bioink optimized for islet encapsulation. They successfully coprinted endothelial progenitor cells and T regulatory cells with islets to produce macroporous 3D constructs composed of coaxially arranged filaments. The coaxial filament provides a protective shell layer and immune protection for the islets located in the core. In addition, the macroporous structure of the 3D bioprinted scaffold increases the surface area to volume ratio and reduces the distance between the islets and the surrounding fluids/blood vessels to less than a few hundred microns. This novel strategy has the potential to address two of the most critical limitations of 3D in vitro models, namely improving viability after islet isolation and promoting revascularization. In addition, this may enable immunoisolation after transplantation [176].

Although they represent a significant improvement over traditional flat culture systems, 3D in vitro models have their own limitations and still face challenges. These include limited access for the functional analysis of entrapped cells, harvesting of organoids and their luminal contents for biochemical and genetic analysis of cells grown on/in scaffolds. Engineered 3D models also cannot yet achieve the full architecture of tissues and tissue–tissue interfaces (e.g., the interface between vascular endothelium, parenchymal cells, and connective tissue), which are critical for tissue/organ function and development. Additionally, cells are not exposed to physiological physicochemical stimuli, such as fluid shear stress, mechanical tension and compression, oxygen tension, and others that affect tissue development and function in health and disease [177].

### 4.4. Bioreactors

The main function of bioreactors is to recapitulate in vivo conditions in which an adequate supply of oxygen, nutrients, and signalling molecules is provided, while waste products are removed to sustain homeostasis. To this end, controlled dynamic cell culture conditions have a clear advantage over static cell cultures. The use of bioreactors is an attractive method of culturing pancreatic cells in vitro, as they continuously perfuse the cells with fresh medium and mimic native physicochemical conditions such as temperature, pressure and pH [178]. To overcome problems related to nutrient and oxygen supply to the cultured islets and improve their viability and functionality, Li et al. developed an in vitro 3D-perfused culture model for diabetic drug efficacy testing. To this end, they encapsulated isolated islets in alginate and cultured them in TissueFlex^®^, a commercially available perfused microbioreactor that can simulate the in vivo perfusion conditions of pancreatic islets. Encapsulated islets that were statically cultured in cell culture plates (3D static) and islets cultured in suspension (2D) were used for comparison. The system supported islet viability and improved functionality in terms of insulin release and dynamic responses over 7 days. The islets showed higher sensitivity in response to two antidiabetic drugs (GLP-1 and tolbutamide) and drug doses compared to conventional 2D and 3D static models [179].

Among the most innovative modern bioreactors are the rotary cell culture system (RCCS) bioreactors developed at National Aeronautics and Space Agency (NASA), which allow for the continuous circular rotation of the cell growth medium. Such culture conditions simulate microgravity and allow cells to be suspended in a “continuous free fall” at terminal velocity, with low hydrodynamic shear stress and turbulence, allowing for a high mass transfer of nutrients and oxygen while maintaining their morphology [180]. Such bioreactors were used by Rutzky et al. to culture pancreatic islets, comparing islets cultured with an RCCS to conventional culture dishes. While fresh islets survived for approximately 15 days when transplanted into streptozotocin-diabetic mice, their counterparts cultured with RCCS survived for over 100 days after transplantation (Figure 6). In addition, a significantly smaller number of RCCS islets were required to maintain euglycemia in the diabetic mice compared to the conventionally cultured islets [181]. This reactor configuration has also been successfully applied to islets derived from human patients suffering from hyperinsulinemic hypoglycemia of infancy (PHHI), and has been shown to provide a unique environment that favours the upregulation of endocrine expression [182].

A study by Hou et al. investigated the effects of dynamic culturing conditions on pancreatic islets under stasis and microgravity, in the presence or absence of a PGA fibrous scaffold, followed by transplantation into diabetic rats. Grafts cultured in simulated microgravity in the presence or absence of a PGA fibrous scaffold were able to significantly normalise insulin production and blood glucose concentration when compared to control grafts. Overall, islets cultured under dynamic conditions showed better viability and increased insulin production compared to those cultured under static conditions. This demonstrates the importance of the scaffold and dynamic culture conditions that must be considered when designing functional in vitro islet models [183].

The main disadvantage of islet culture in these bioreactors is the tendency of the cultured islets to aggregate when cultured at higher densities, which increases the risk of anoxic necrosis. However, one study has shown that human islets cultured at high densities of 500–1500 islets/mL in a rotating cell culture system with a configuration known as High Aspect Ratio Vessel (HARV) exhibit less central necrosis and maintain their structural integrity and glucose-stimulated insulin release over a 10-day culture period due to highly efficient gas exchange [184]. Furthermore, human islet culture has been shown to be supported by the coculture of epithelial ductal cells under rotational conditions [185].

Although this technology has not been widely explored, it has shown great promise for culturing and maintaining islet function in vitro. Further improvements and experiments with this technology, such as the use of solid supports for islet culture, namely microcapsules and scaffolds, are needed to further improve islet culturing techniques in vitro [178].

### 4.5. Pancreas-on-a-Chip

Organs-on-a-chip, also known as tissue chips, are microfluidic (microphysiological) cell culture devices manufactured using microchip fabrication techniques (Figure 7) that allow for the controlled design of surface “features” in the nm–μm range [186]. Microfluidic devices allow for precise control over fluid mixing, the establishment of stable gradients, controlled shear forces, etc. [186,187]. The chips can be inhabited by living cells and can also contain continuously perfused micrometer-sized chambers arranged to recapitulate tissue- or organ-level physiology and allow cells to respond to hormones, cell signalling molecules, biomechanical stimuli, and/or drugs. The designed surfaces allow cells to sense and respond to various stimuli, similar to their natural microenvironment, partially mimicking functional tissue units [188].

Depending on the design, microfluidic systems can self-assemble cells into spheroids [189] or organoids [190], cell positioning within defined locations within scaffolds [191], etc. The possibilities are numerous, ranging from simple 2D cell culture within a perfused microfluidic chamber to interconnected chips with different in vitro organ models mimicking whole-body systems [4]. Microfluidic chips allow for control over parameters (e.g., flow rate, shear stress, pressure) that are difficult to regulate in static 3D in vitro models or bioreactors, and can be coupled with microsensors that provide feedback on cells or microenvironment conditions, which is not typically possible in standard 3D models, creating complex physicochemical gradients [192]. As these devices simultaneously mimic native tissue architecture, cell–cell and cell–ECM interactions, tissue–tissue interfaces, physicochemical cues, and vascular perfusion of the body, they replicate tissue- and organ-level functionality at a level not possible with previously described conventional 2D and 3D in vitro models [193]. Moreover, these microchips enable high-resolution, real-time imaging and in vitro analysis of biochemical, genetic, and metabolic processes in living cells in the context of functional tissues and organs. Therefore, they represent a significant advance in the study of organ and tissue physiology, development, and disease etiology. Although their use is increasing, they are still most commonly used in the context of drug discovery and development, as they provide the basis for improved studies of molecular mechanisms of drug activity, toxicity testing, and prioritization of lead candidates [194].

To date, research using pancreatic microfluidic systems has focused on culturing islets of Langerhans to measure their functions prior to transplantation [195]; deterioration of islet EC morphology due to sheer stress and media composition [196]; investigation of the effects of flow velocity around the islets and media flow through the tissue on islet morphology and functionality and coupled ECs [67]; insulin production under defined temporal glucose gradients [197]; a real-time insulin detection system based on a rapid electrophoresis-based immunoassay [198,199]; and a method to monitor islet cell functionality in microfluidics by measuring zinc secretion [200], as Zn^2+^ is co-secreted by beta cells with insulin [201]. For relevant reviews on the applications of microfluidics technology for pancreatic islet research, we refer the reader to the reviews by Wang et al. [202] and Castiello et al. [203].

More recently, for in vitro modelling applications, Nguyen et al. developed an endocrine system on chip for a diabetes treatment model to monitor dynamic changes in endocrine hormones (GLP-1 and insulin) in a glucose-dependent environment. To do this, they constructed a co-culture of intestinal pancreatic cells to measure the effect of glucose on the production of GLP-1 from the L-cell line (GLUTag) and insulin from the pancreatic β-cell line (INS-1). After three days of culture, both cell lines formed aggregates that exhibited 3D cell morphology and high viability (>95%). They separately measured the dynamic profile of GLP-1 and insulin release at glucose concentrations of 0.5 and 20 mM, and the combined effect of GLP-1 on insulin production at these glucose concentrations. In response to glucose stimuli, GLUTag and INS-1 cells produced higher amounts of GLP-1 and insulin, respectively, compared to a static 2D cell culture. INS-1 combined with GLUTag cells showed even higher insulin production in response to glucose stimulation. At higher glucose concentrations, the diabetes model on chip showed faster saturation of insulin levels [204]. Conversely, Bauer et al. developed a microfluidic two-organ chip that can serve as a type 2 DM in vitro disease model. To this end, they co-cultured human liver spheroids, consisting of HepaRG cells and primary human stellate cells, with human pancreatic islet microtissues to model their physiological cross-talk. Functional coupling, evidenced by the release of insulin from the islets in response to a glucose load in glucose tolerance assays on different days, promoted glucose uptake by liver spheroids. Co-cultures maintained postprandial glucose concentrations in the circulation while glucose levels remained elevated in both single cultures. Thus, insulin secreted into the circulation stimulated glucose uptake by liver spheroids, whereas the latter did not consume glucose as efficiently in the absence of insulin. When glucose concentration decreased, insulin secretion decreased, demonstrating a functional feedback loop between the liver and insulin-secreting islet microtissues [205].

Both previously mentioned studies used microfluidic systems fabricated from PDMS, which requires specific photolithographic approaches. Furthermore, the inherent size limitations of this fabrication method pose a challenge in the development of microphysiological systems for organoid culture. In a recent study, Patel et al. addressed these shortcomings and were the first to embed pancreatic islets in an alginate hydrogel and integrate them into a PDMS-free microfluidic system. The effects of the culture systems on islet health were evaluated and compared both in silico and in vitro. Computational modelling predicted reduced hypoxic stress and improved insulin secretion compared to static culture. Experimental validation by serial, high-content, and non-invasive assays quantitatively confirmed that the platform maintained organoid viability and functionality for at least 10 days, in stark contrast to the acute decline observed overnight under static conditions. With this study, the group was also the first to demonstrate a serial, dynamic assessment of the same set of islets over longer culture periods, with both offline functional assessment of dynamic glucose-stimulated insulin secretion and simultaneous in situ tracking of viability and calcium signalling [206].

Overall, microphysiological systems demonstrate the importance of a dynamic in vitro microenvironment for the maintenance of primary organoid function and highlight their ability to elucidate the complex physiological and pathophysiological processes of islets by combining dynamic 3D culture, optical assessment and functional assays. Furthermore, such platforms can be used to study temporal interactions between complex matrices and other tissues while limiting the amount of sample required for assessment. Ultimately, the coupling of microphysiological culture conditions and multiparametric assessment will provide valuable data for the development of improved in vitro disease models, novel therapeutics, and the study of diabetes.

### 4.6. Tissue Slices

Due to the many limitations and complexity of constructing relevant in vitro models of the pancreas, as this organ has a complex structure and function [29], isolated endocrine cells, especially beta cells, isolated acinar cells, isolated islets of Langerhans, and isolated ducts and acini, are still most commonly used for basic and translational studies of the pancreas [208]. The results of these studies are often complemented by various in vivo measurements at the whole-organism level [209].

The recently developed and increasingly used method of pancreatic tissue slice, is an important step towards the development of the best possible 3D model for pancreatic tissue, which is in many ways an alternative, and in some cases a complement, to the development of advanced 3D models [208,210]. In this method, pancreatic tissue from various model organisms (usually mice, but also rats or pigs) or humans is cut into approximately 100 µm thick tissue slices. The tissue slices contain intact islets of Langerhans, sliced islets of Langerhans, large areas of intact acini, and long sections of ducts of various sizes. One of the most important features of the tissue slices thus obtained is that the cells within the islets of Langerhans and within the acini are connected by various intercellular contacts, thus preserving paracrine interactions within the endocrine part, within the exocrine part, and between the two parts. To a greater extent than isolation of the islets and acini, the vessels, basement membrane, connective tissue envelopes, immune cells, and other elements of the mesenchyme are also preserved, thus also preserving the 3D structure of the tissue [208,211,212,213]. Acute pancreatic slices are, therefore, a special form of primary cell cultures that can be used for at least 24–48 h without special additional scaffolds [212,214].

In combination with electrophysiological measurements, intracellular calcium ion concentration measurements, secretion measurements and various morphological measurements, the tissue slice method has been shown to be at least equivalent to cell and islet or acinar isolation methods in terms of results and repeatability [33,215,216,217,218,219,220]. However, in many respects, tissue slice allows for more physiological data, especially when assessing communication between different cells [66,221,222,223]. Another important advantage of the tissue slice method is that it is compatible with many of the different model organisms with fluorescently labelled cells in which we are interested. At the same time, it is also compatible with diabetes models that lead to the destruction of most beta cells (e.g., streptozotocin model) and, therefore, not compatible with the isolation of cells or islets, since in these cases too little isolate is obtained [224,225]. The results of morphological and functional measurements in the slice are becoming the gold standard in this field and an important reference for measurements in other 3D models.

Finally, it is worth highlighting that tissue slices have recently been used in combination with special scaffolds that extend the life of the ex vivo slice and its usability for studies by at least one week [212]. When using human tissue, it is expected that the scaffolds will need to allow for tissue transfer for a duration of several days in order to be used for studies in parts of the world where there is no local source of human tissue, or for measurements in specialized laboratories using methods that are not available at the site of the human tissue source. We anticipate that a larger or smaller portion of the tissue from the slice may also be used in the future to build 3D models using the advanced materials and techniques mention earlier, combining cells with these materials [167].

## 5. Conclusions

The highly interdisciplinary field of in vitro modelling is rapidly evolving and advancing, but despite the already exciting breakthroughs in the design, fabrication, and validation of tissue/organ models with increased complexity, there are still challenges to overcome. Currently, the most advanced 3D in vitro models of the pancreas mimic only the basic functions of exo- or endocrine tissue, so, despite the shortcomings, tissue slices are used as the gold standard for performing basic studies. As discussed in this review, one of the many limitations of such models begins early in their design, namely, at the stage of identifying the right cell sources for model development. Second, it is particularly difficult to isolate and culture human primary cells in vitro for extended periods of time (regardless of tissue and cell type), as primary cultures are very sensitive to culture conditions and progressively de-differentiate after only a few passages. In the last decade, the most promising breakthrough in this field is the discovery of iPSCs. The reprogramming of adult cells into embryonic-like states has a myriad of potential applications in regenerative medicine. Although there are many basic and technical scientific concerns, iPSCs represent an interesting cell source and promise to open many new possibilities in the field of in vitro modelling. The immature phenotype exhibited by differentiated progenitor-derived cells makes them useful for early-stage disease models. In addition, the properties of human cells are highly dependent on the tissue source, the health and age of the donor, the isolation/purification method used, and the differentiation protocol applied when iPSCs are involved.

The intricate 3D microenvironment surrounding the cells exerts a synergistic role in controlling cell behavior. Therefore, it is essential to mimic the architecture, biochemical and mechanical properties, as well as the dynamic environmental conditions of the in vivo environment, as closely as possible in vitro. The development of a biomimetic microenvironment is crucial for the long-term culturing of any cell type. This is particularly challenging due to the complexity of human tissues/organs and the difficulty of recapitulating all physiological cues, mechanical, topographical and chemical aspects at different stages of health and age. In this regard, the combination of new advances in microfabrication technology, biomaterial science and microfluidics is becoming increasingly important. Advances in biomaterial science, including the design and development of new natural and synthetic copolymers or bioartificial blends, must be exploited to fine-tune the mechanical, chemical, thermal, and surface properties of scaffold-forming materials [226,227]. Further development of tailored materials would allow the bulk properties of native healthy or pathological tissues to be mimicked. In addition, new advances in scaffold fabrication are gaining interest as they enable the fabrication of reproducible scaffolds with good control over nano-, micro- and macro-properties (e.g., porosity) that facilitate nutrient delivery, gas diffusion and waste removal. Emerging rapid prototyping techniques, bioprinting, and bottom-up approaches are promising tools, with the potential to overcome the drawbacks of conventional approaches due to their high versatility and spatiotemporal control [135,228]. Finally, to enable high-throughput applications and increase model scalability, recent advances in microfluidics, combined with tissue engineering approaches, are now being integrated into perfused organ-on-a-chip models, which may be key to the development of more relevant in vitro models of the pancreas [192].

## Figures and Tables

**Figure 1 biomedicines-09-01415-f001:**
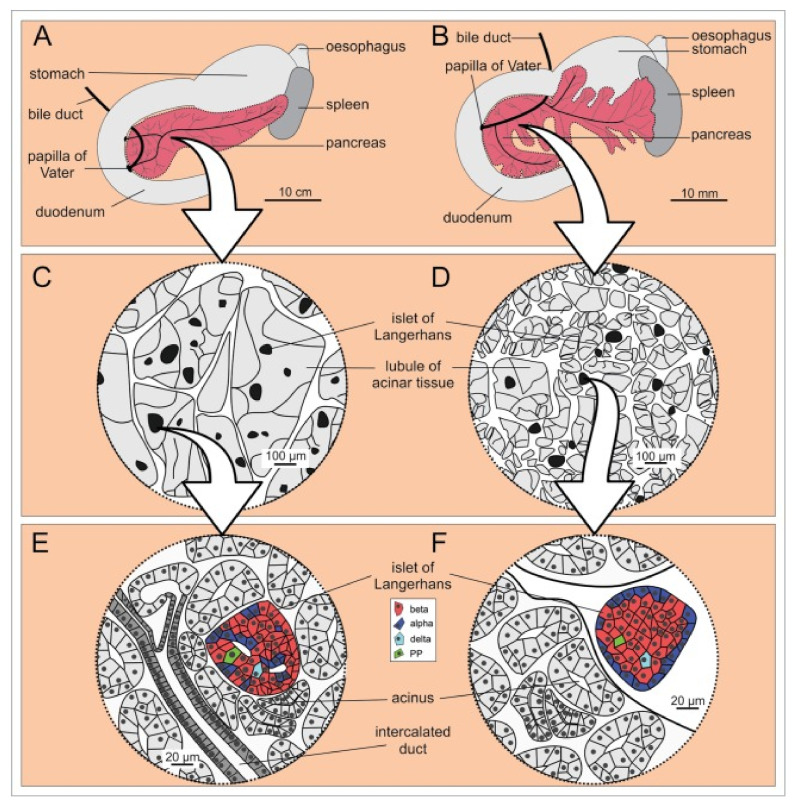
Microscopic anatomy of human (**A**,**C**,**E**) and mouse (**B**,**D**,**F**) pancreas. (**A**,**B**) Macroscopic anatomy of human and mouse pancreas, respectively. (**C**,**D**) Enlargement of a portion of the pancreas shows that the lobules are larger in humans than in mice, whereas the islets of Langerhans are of comparable size in humans and mice. (**E**,**F**) The cell composition and location of the islets of Langerhans within the pancreas are markedly different in the two species. Note the more diffusely distributed endocrine cells in humans (**E**) and the mantle-core pattern in mice (**F**) [29]. Reprinted (adapted) from Creative Commons Attribution License CC BY 3.0.

**Figure 2 biomedicines-09-01415-f002:**
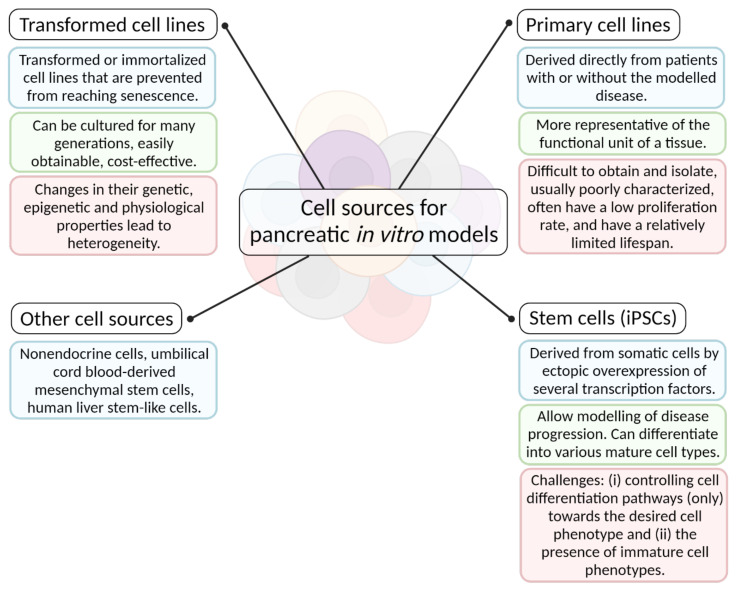
A diagram of various cell types used for in vitro disease models of the pancreas, showing their advantages and disadvantages (created with BioRender.com; accessed on 6 July 2021).

**Figure 3 biomedicines-09-01415-f003:**
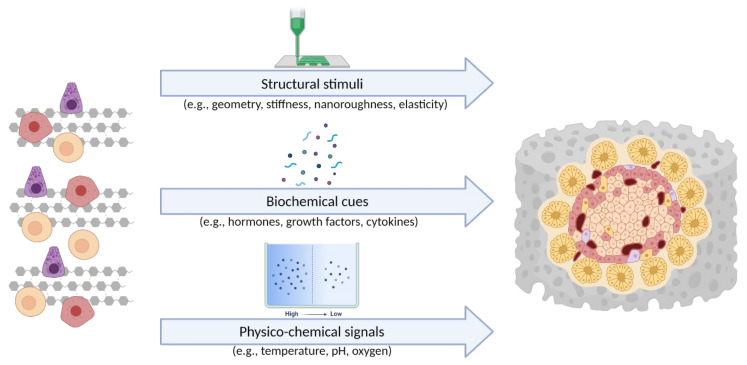
The three groups of cues, which act on cells in a 3D in vitro model [105] (created with BioRender.com; accessed on 6 July 2021).

**Figure 4 biomedicines-09-01415-f004:**
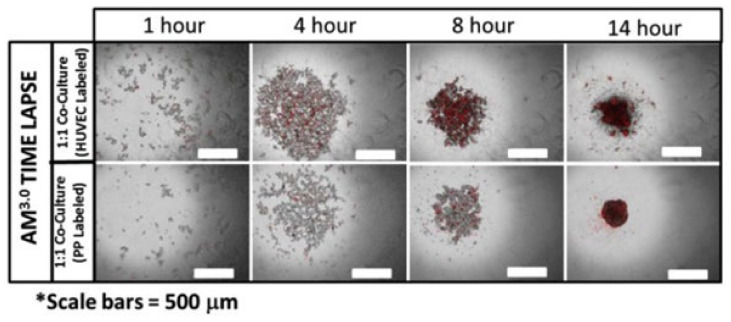
Generation of heterogeneous, multicellular islet organoids on Amikagel. Time-lapse imaging of co-culture aggregation on Amikagel (labeled cell population in red: HUVEC top row and hESC-PP bottom row). Reprinted with permission from [115]. Copyright (2018) Elsevier.

**Figure 5 biomedicines-09-01415-f005:**
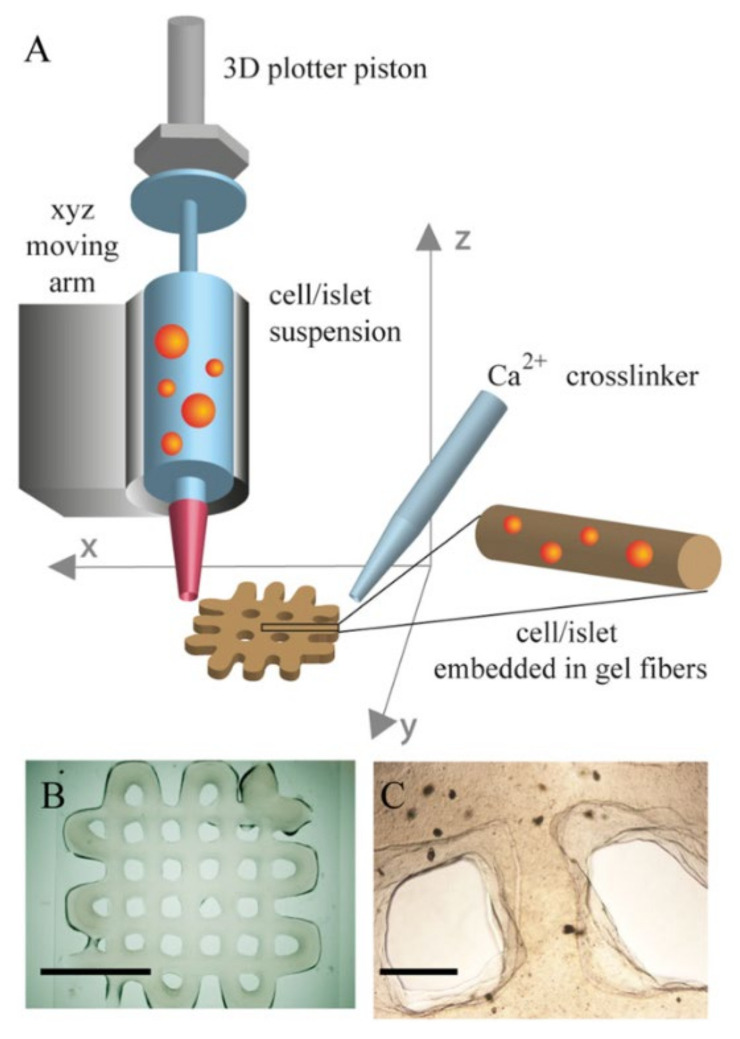
3D bioprinting scaffolds. (**A**) 3D printing schematic. (**B**) alginate/gelatin 2 × 2 cm^2^ plotted scaffold (scale bar 10 mm). (**C**) Image of plotted islets in the construct (scale bar 1 mm). Reprinted with permission from [164]. Copyright (2015) IOP Publishing.

**Figure 6 biomedicines-09-01415-f006:**
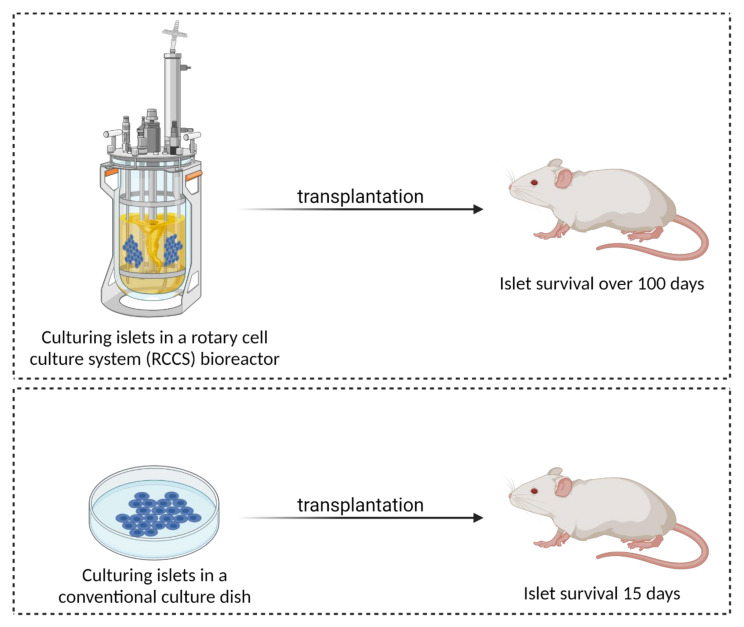
A comparison of conventional culture dish culturing and culturing in a rotary cell culture system (RCCS) bioreactor (created with BioRender.com; accessed on 6 July 2021).

**Figure 7 biomedicines-09-01415-f007:**
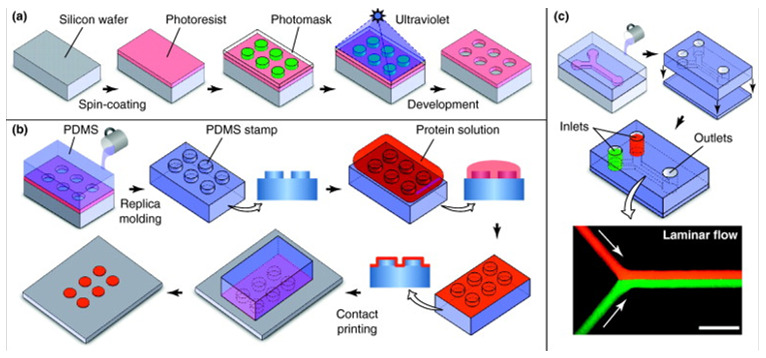
Microengineering technologies used to construct 3D culture systems and organs-on-chips. (**a**) Photolithography, (**b**) soft lithography, (**c**) microfluidic devices. Reprinted with permission from [207]. Copyright (2011) Elsevier.

**Table 1 biomedicines-09-01415-t001:** Summary of different types of in vitro cell models categorized based on their complexity.

In vitro model	Description	Advantages	Disadvantages
Monolayer monocultures	Cells are grown on the flat surface of a plastic cell culture container and covered with culturing medium.	Ease of use, low cost, high throughput, high reproducibility, standardization.	Non-physiological structural support (stiff, nonpermeable, 2D substrate) and cellular alignment resulting in loss of phenotype, changes cell shape. Static environmental conditions.
Monolayer monocultures with ECM	Surface of the cell culture container is coated using ECM or ECM substitutes on which the cells are cultured.	Cost-efficient. Mimic some aspects of cell-ECM interaction, e.g., allows for the modulation of cell behaviour and signalling. Especially useful for epithelial cultures.	Loss of phenotype, lack of more complex 3D architecture, altered cell shape. Static environmental conditions.
Monolayer co-cultures	Monolayer cultures containing more than one cell type.	Ease of use, low cost, allows for the study of interactions between cell types, synergistic/antagonistic effects of medium consumption and metabolite production.	Difficult to analyse the contribution of each cell type to co-culture function. Depending on cell growth rate, one cell type can rapidly dominate the culturing container. Static environmental conditions.
3D cultures (organoids)	Cells grown in a 3D environment. The 3D environment can be created by an ECM or by a nonadherent surface.	Cost efficient, ease of use. High cell density, mimics the 3D architecture of native tissues. Prolonged cell viability and functionality. Self-assembly.	Limited by nutrient/oxygen diffusion allowing for only small sample sizes. Difficult to functionally analyse entrapped cells. Limited control over shape/size. Static environmental conditions.
3D cultures (scaffolds)	Cells grown on artificial supports (scaffolds) which mimic the native ECM	Improved nutrient/oxygen diffusion. Mimics the 3D architecture of native tissues. Prolonged cell viability and functionality. Adjustable physico-chemical properties and their gradients.	Complex design and manufacturing procedures. Difficult to functionally analyse entrapped cells. Limited range of biocompatible materials. Static environmental conditions.
3D co-cultures (spheroids, scaffolds)	More than one cell type is grown in a 3D environment.	Reproduce important functions of the target tissue/organ. Prolonged cell viability and functionality.	Complex/costly. Difficult to functionally analyse entrapped cells. Static environmental conditions.
Bioreactors	Cells grown in reactors that mimic physiological conditions	Simulate “micro gravity”, mimic physicochemical gradients, constant perfusion. High throughput.	Homogenization of cell-medium suspension does not allow for implementation of chemical and structural gradients. Shear stress. Cells tend to aggregate which leads to central necrosis. Costly.
Organ-on-a-chip	A microfluidic cell culture device containing continuously perfused chambers inhabited by living cells.	Simulate tissue- and organ-level physiology. Sensor integration and real-time measurement of multiple analytes.	Complex, multistep fabrication methods. Costly with low throughput.
Tissue slices	Tissue slices are cultured in plates or on supports.	In vivo environment is preserved.	Requires hyperoxic conditions. Cell viability limited over short periods of time.

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
