# Peer review of "In Vitro Disease Models of the Endocrine Pancreas"

_biomedicines, 2021, doi:10.3390/biomedicines9101415_

Round 1

Reviewer 1 Report

In this review the authors summarize the current knowledge regarding the in vitro disease models of the pancreas. After the brief introduction about the need of these in vitro models they introduce the organ itself and highlight the differences between human and mouse pancreata. The major part of the review consists of the detailed description of the different types of in vitro cell models. The review is very comprehensive, however it focuses mainly on the endocrine part, especially on Langerhans cells and there is very limited information regarding the acinar and ductal cells and their in vitro models – as the title is „In vitro disease models of the pancreas“, I would expect this. I would suggest to focus not only on diabetes but also on pancreatitis and pancreatic cancer, as these are the major disorders of the pancreas.

Minor: The numbering of the subtitles is little bit confusing. The different types of in vitro models of the pancreas should be designated as 4.1., 4.2., etc.

Author Response

We thank the reviewer for their comments and constructive criticism. The observation that the manuscript focuses primarily on the endocrine pancreas is correct. This was in fact our intention, but seems to have been poorly conveyed. To amend this, we have changed the title and modified the abstract and main text of the manuscript to include the following text:

“Research focusing on pancreatic in vitro models has already been described and assembled in some excellent reviews, especially those focusing on pancreatic cancer [13-17], but also pancreatitis [18]. With this in mind, this review provides an overview of in vitro tissue and disease models with a focus on in vitro disease models of the endocrine pancreas.”

We have also changed the numbering of the subtitles according to the reviewers’ suggestions.

Reviewer 2 Report

The authors have undertaken a fairly comprehensive review of in vitro models of the islets of Langerhans and beta cells.  The authors concluded that the highly interdisciplinary field of in vitro modelling is rapidly evolving and advancing, however, there are still challenges to overcome.  I have some comments that I believe need to be addressed prior to publication of this article.

Page 6 lines 213–134, “In addition, they lose critical ECM and basement membrane contacts.”, Please define ECM.

Page 6 lines 226–228, “Interestingly, culturing isolated islets in a microfluidic media flow results in higher cell density and better connections with surrounding cells (e.g. ECs) compared to islets cultured in conventional static 2D cultures.”, Please define ECs here.

Page 6 lines 230–232, “However, high media flux can lead to attenuated glucose-stimulated metabolic and Ca2+ responses of beta cells at the periphery of islets, consistent with shear-induced damage [87].”, References 87 is inappropriate in this context.

Author Response

We thank the reviewer for pointing out the flaws and errors in the manuscript. We have corrected the raised points and hope that the revised text is appropriate. The pointed out reference was placed there by mistake and has now also been revised.

Round 2

Reviewer 1 Report

The authors changed the title and clarified that their intention was to summarize the in vitro disease models of endocrine pancreas.